# Association of Functional Gastrointestinal Disorders with Adolescent Idiopathic Scoliosis

**DOI:** 10.3390/children11010118

**Published:** 2024-01-18

**Authors:** Soo-Bin Lee, Hyun-Wook Chae, Ji-Won Kwon, Sahyun Sung, Seong-Hwan Moon, Kyung-Soo Suk, Hak-Sun Kim, Si-Young Park, Byung Ho Lee

**Affiliations:** 1Department of Orthopedic Surgery, Catholic Kwandong University International St. Mary’s Hospital, Incheon 22711, Republic of Korea; sumanzzz@ish.ac.kr; 2Department of Pediatrics, Yonsei University College of Medicine, Seoul 03722, Republic of Korea; hopechae@yuhs.ac; 3Department of Orthopedic Surgery, Yonsei University College of Medicine, Seoul 03722, Republic of Korea; kwonjjanng@yuhs.ac (J.-W.K.); shmoon@yuhs.ac (S.-H.M.); sks111@yuhs.ac (K.-S.S.); haksunkim@yuhs.ac (H.-S.K.); drspine90@yuhs.ac (S.-Y.P.); 4Department of Orthopedic Surgery, Ewha Womans University Seoul Hospital, Seoul 07804, Republic of Korea; sahyunsung@ewha.ac.kr

**Keywords:** functional gastrointestinal disorder, idiopathic scoliosis, functional dyspepsia, irritable bowel syndrome

## Abstract

Numerous adolescents diagnosed with adolescent idiopathic scoliosis (AIS) often manifest symptoms indicative of functional gastrointestinal disorders (FGIDs). However, the precise connection between FGIDs and AIS remains unclear. The study involved adolescents drawn from sample datasets provided by the Korean Health Insurance Review and Assessment Service spanning from 2012 to 2016, with a median dataset size of 1,446,632 patients. The AIS group consisted of individuals aged 10 to 19 with diagnostic codes for AIS, while the control group consisted of those without AIS diagnostic codes. The median prevalence of FGIDs in adolescents with AIS from 2012 to 2016 was 24%. When accounting for confounding factors, the analysis revealed that adolescents with AIS were consistently more prone to experiencing FGIDs each year (2012: adjusted odds ratio (aOR), 1.21 [95% confidence interval (CI), 1.10–1.35], *p* < 0.001; 2013: aOR, 1.31 [95% CI, 1.18–1.46], *p* < 0.001; 2014: aOR, 1.24 [95% CI, 1.12–1.38], *p* < 0.001; 2015: aOR, 1.34 [95% CI, 1.21–1.49], *p* < 0.001; and 2016: aOR, 1.35 [95% CI, 1.21–1.50], *p* < 0.001). These findings suggest that AIS is correlated with an elevated likelihood of FGIDs, indicating that AIS may function as a potential risk factor for these gastrointestinal issues. Consequently, it is recommended to provide counseling to adolescents with AIS, alerting them to the heightened probability of experiencing chronic gastrointestinal symptoms.

## 1. Introduction

Among pediatric spinal deformities, adolescent idiopathic scoliosis (AIS) stands out as the most prevalent type. If a spinal curvature of Cobb angle 10 degrees or more is observed between the ages of 10 and 19 without a specific identifiable cause, it can be diagnosed as AIS [1]. The exact cause of AIS remains unknown, but genetic factors, neurologic disorders, hormonal dysfunctions, and environmental factors have been proposed as etiologic factors. Patients with AIS may exhibit physical changes such as deformity of the torso due to the spinal curvature, uneven shoulder height, and prominence of the ribs. The severity of these changes tends to worsen with the degree of curvature. Chronic lower back pain may accompany AIS, and in severe cases with a Cobb angle exceeding 70 degrees, there can be a decline in lung function leading to symptoms such as carbon dioxide retention, pulmonary hypertension, and cor pulmonale. In these severe AIS cases, the risk of premature death is more than twice as high as in the general population. Spontaneous improvement is rare, and curve progression is often observed during rapid adolescent growth. The factors associated with the progression of curvature are crucial, with age and the magnitude of the curvature playing significant roles [2]. The younger the age and the larger the curvature, the greater the risk of curvature progression. Observation and bracing are the most widely accepted nonoperative treatments for mild to moderate curves, whereas operative treatment is required for Cobb angles of 50 degrees or more [3,4]. Various surgical methods exist, but posterior spinal fusion surgery using pedicle screws is commonly and widely performed. Recently, minimally invasive surgical techniques have also gained attention [5].

Functional gastrointestinal disorders (FGIDs) are disorders of gut–brain interaction. They are characterized by chronic gastrointestinal symptoms, such as abdominal pain, dyspepsia, and diarrhea, in the absence of demonstrable pathology on conventional testing [6]. FGIDs exhibit a high prevalence worldwide, regardless of geographical location, and are particularly recognized as the most prevalent disorders among patients within the digestive system. FGIDs are an important issue, with annual costs of up to USD 358 million in the United States and EUR 8 billion in Europe [7,8]. The exact causes of FGIDs are not precisely understood. It is speculated that a combination of factors such as disturbances in sensory, motor, and absorptive functions of the gut, past gastrointestinal infections, and psychosocial factors may play a complex role. Recently, there is growing consideration that abnormalities in the brain–gut axis, representing a complex network of bidirectional communication between the central and enteric nervous systems, play a significant role in the onset of FGIDs.

Functional dyspepsia and irritable bowel syndrome are representative examples of FGIDs. Major symptoms of functional dyspepsia include upper abdominal pain, discomfort, bloating, and nausea, while key symptoms of irritable bowel syndrome encompass abdominal pain, constipation, diarrhea, and mucous stool. If an individual complains of digestive symptoms for more than three months without any apparent underlying cause, and no specific disease is evident, a diagnosis of FGID can be considered. Various tests, including endoscopy and blood tests, may be conducted; however, there are no specific diagnostic tests for FGIDs, as the results often appear normal. In cases of severe symptoms, pharmacological treatment may be administered to alleviate the condition. Moreover, as FGIDs arise from complex bidirectional dysregulations of gut–brain interaction, management emphasizes biopsychosocial approaches, including lifestyle and diet modifications, along with the treatment of coexisting psychological comorbidities [9]. Despite such treatments, FGIDs often exhibit a chronic course. FGIDs may be associated with a reduced quality of life, similar to the poor quality seen with other chronic medical conditions, such as congestive heart failure and rheumatoid arthritis [10].

In adolescent patients displaying gastrointestinal symptoms without a clear identifiable cause, there are instances where scoliosis is concomitant. Conversely, patients with scoliosis may also exhibit functional gastrointestinal symptoms. Several studies have addressed the reduced quality of life and psychosocial characteristics of adolescents with AIS, which may be the result of cosmetic concerns and poor self-image [11,12,13,14]. In our previous study, we observed a higher prevalence of psychiatric disorders in adolescents with AIS than in adolescents without AIS [15]. Considering the causal relationship between psychological distress and FGIDs, we hypothesized that there may be a link between FGIDs and AIS. To the best of our knowledge, no previous study has examined the association between FGIDs and AIS in adolescents. In this research, we examined the prevalence of FGIDs in adolescents diagnosed with AIS and assessed the correlation between FGIDs and AIS by analyzing data from the Korean National Health Insurance database.

## 2. Materials and Methods

Ethics approval for this study was obtained from the Institutional Review Board (IRB) of the corresponding author’s hospital (Yonsei University Gangnam Severance Hospital IRB and Ethics Committee: 3-2018-0041). All methods were performed in accordance with the Declaration of Helsinki and Yonsei University’s institutional guidelines.

### 2.1. Data Source

This research utilized patient information sourced from the Korean Health Insurance Review and Assessment Service (HIRA). All citizens in Korea are covered by national health insurance or medical aid. HIRA conducts reviews and assessments of medical fees when health institutions file a claim for medical fees to HIRA for reimbursement. HIRA collects patient data, including age, gender, diagnoses, and prescriptions. After anonymization and random extraction, statistically sampled datasets are available for researchers. The datasets used in this study were HIRA-NPS-2012-0133, 2013-0143, 2014-0152, 2015-0151, and 2016-0050, covering the years 2012 through 2016. The results of this study reflect our analysis of the provided data, independent of HIRA or the Ministry of Health and Welfare of Korea.

### 2.2. Study Population

We analyzed HIRA datasets from 2012 to 2016. Each dataset was a 3% randomly extracted sample of the total number of patients within each year. All patients in the HIRA datasets who visited medical institutes for any kind of disease or symptoms were included. The median number of all patients in the datasets was 1,446,632 (range, 1,421,707–1,468,033). The exclusion criterion was set based on age, and patients under the age of 10 and those over 19 years old were excluded. After applying the age criteria, the median number of adolescents analyzed in the 2012–2016 datasets was 177,630 (range, 165,647–188,780).

The following diagnostic codes for scoliosis were used for AIS group, as defined by the International Classification of Diseases, version 10 (ICD-10): M41.2 (other idiopathic scoliosis), M41.3 (thoracogenic scoliosis), M41.8 (other forms of scoliosis), and M41.9 (scoliosis, unspecified). Adolescent patients aged 10 to 19 with the diagnosis code for scoliosis mentioned above were classified into the AIS group. On the other hand, adolescent patients aged 10 to 19 without the mentioned scoliosis diagnosis code were classified into the control group. Unlike AIS, neuromuscular scoliosis, which can be caused by conditions such as cerebral palsy or spinal muscular atrophy, or secondary scoliosis resulting from trauma, have a distinct clinical presentation. The limitation of movement associated with these conditions may influence the occurrence of gastrointestinal diseases. Therefore, we excluded diagnostic codes for other types of scoliosis, including M41.0 (infantile idiopathic scoliosis), M41.1 (juvenile idiopathic scoliosis), M41.4 (neuromuscular scoliosis), M41.5 (other secondary scoliosis), and Q76.3-4 (congenital scoliosis).

The number of patients diagnosed with an FGID was determined in each group, using these ICD-10 diagnostic codes for FGIDs: K30 (functional dyspepsia), K58 (irritable bowel syndrome), and K59 (other FGIDs). A flow diagram of the inclusion and exclusion criteria of the patients included in this study is shown in Figure 1.

### 2.3. Primary Study Outcome

The main focus of the study was to examine the connection between FGIDs and AIS, while accounting for potential confounding factors. The calculation of patient numbers relied on diagnostic codes. The study assessed the percentage of adolescents with AIS who also exhibited FGIDs over the five years under investigation. Age, gender, insurance type, and residual district codes were chosen as variables for the multivariable analysis, and adjusted odds ratios (aORs) were computed for each year.

### 2.4. Statistical Analysis

Relevant clinical variables were chosen from a diverse set of codes within the dataset. Each variable underwent univariable logistic regression analysis, and a significance threshold of *p* < 0.05 was employed to progress a factor to multivariable analysis. Subsequently, multivariable logistic regression analysis was conducted to ascertain the aOR for FGIDs in adolescents diagnosed with AIS. A fixed-effect meta-analysis was conducted for the 5-year multivariable analysis results, and the overall odds ratio for FGIDs in adolescents with AIS was obtained. All statistical analyses were performed using SAS version 9.4 software (SAS Institute, Cary, NC, USA) and RStudio version 4.3.1 (RStudio Team, Boston, MA, USA).

## 3. Results

The median number of total adolescents analyzed in the datasets from 2012 to 2016 was 177,630 (range, 165,647–188,780). The median average age was 14.6 (range, 14.4–14.6) years, and the median proportion of girls was 48.6% (range, 48.4–48.7%).

The overall prevalence of AIS among adolescents in the datasets from 2012 to 2016 was 1.1%. The median number of adolescents with AIS in the datasets from 2012 to 2016 was 1918 (range, 1851–2023). The median average age was 14.8 (range, 14.6–14.9) years, and the median proportion of girls was 62.0% (range, 61.4–63.4%). The median number of adolescents in the control group was 175,626 (range, 163,796–186,757). The median average age was 14.6 (range, 14.4–14.6) years, and the median proportion of girls was 48.4% (range, 48.2–48.5%).

During the 5-year study period, the median prevalence of FGIDs was 23.9% (range, 23.0–24.9%) among adolescents with AIS and 19.1% (range, 18.5–21.0%) among adolescents in the age-matched control group (Table 1).

After adjusting for age, gender, insurance type, and residual district, adolescents with AIS were more likely to have an FGID in each of the 5 years, compared with adolescents without AIS. The aORs for FGID in adolescents with AIS, compared with adolescents without AIS, ranged from 1.21 to 1.35 (2012: aOR, 1.21 [95% confidence interval (CI), 1.10–1.35], *p* < 0.001; 2013: aOR, 1.31 [95% CI, 1.18–1.46], *p* < 0.001; 2014: aOR, 1.24 [95% CI, 1.12–1.38], *p* < 0.001; 2015: aOR, 1.34 [95% CI, 1.21–1.49], *p* < 0.001; and 2016: aOR, 1.35 [95% CI, 1.21–1.50], *p* < 0.001) (Table 1). Multivariable logistic regression analysis for the 2015 data is shown in Table 2 as an example. All additional detailed analyses from 2012 to 2016 can be found in the Appendix A. The fixed-effect meta-analysis of the multivariable logistic regression results over the 5-year period showed that the overall odds ratio for FGIDs in adolescents with AIS was 1.29 (95% CI, 1.23–1.35). The forest plot depicting the results of the meta-analysis is shown in Figure 2.

## 4. Discussion

FGIDs are common among adolescents and may cause a significant burden to both individuals and society. However, to the best of our knowledge, no previously published study has examined the prevalence of FGIDs among adolescents with AIS and whether there is an association between FGIDs and AIS. In this study, we found that approximately 24% of adolescents with AIS had an FGID. We also detected an association between FGIDs and AIS. After controlling for age, gender, insurance type, and residual district, adolescents with AIS had increased odds of FGIDs, compared with the control group. Hence, spinal surgeons should be cognizant of the possibility of FGIDs in adolescents with AIS and consider early referral to pediatricians when gastrointestinal symptoms are present. 

Adolescents with FGIDs present with various gastrointestinal symptoms, such as abdominal pain, nausea, vomiting, constipation, and bowel incontinence, which cannot be attributed to another medical condition after appropriate diagnostic testing. FGIDs in adolescents may lead to a reduced quality of life, school absenteeism, missed work for parents, and greater health care expenditures [16]. In a 2017 systematic review, the prevalence of FGIDs among adolescents was 9.9–29% in school samples and up to 87% in clinical samples [17]. Currently, there are no specific biomarkers or “gold standard” tests for diagnosing FGIDs. The Rome IV criteria [18] are guidelines based on a detailed clinical evaluation and are widely used in pediatric practice. Because diagnostic methods are based on symptoms, most existing epidemiologic studies have used questionnaires. In a study of 864 schoolchildren and adolescents, the overall prevalence of FGIDs was 30% [19]. In another study, Scarpato et al. reported an overall 26.6% prevalence of FGIDs among 7148 adolescents in the Mediterranean region of Europe [20]. In the current study, the overall prevalence of FGIDs was 23.9% in adolescents with AIS and 19.1% in adolescents without AIS over the 5-year study period. As our findings relied on diagnostic codes recorded in a medical registry, we are confident that our results accurately represent the actual prevalence of FGIDs in adolescents in real-world scenarios.

The pathophysiology of FGIDs is explained by the biopsychosocial model developed by Engel [21] and adapted by Drossman [22]. For decades, alterations in gastrointestinal motility, visceral hypersensitivity, and psychological disturbances have been recognized as contributors to the pathogenesis of FGIDs [23]. Limitation of movement was also suggested to be associated with the occurrence of FGIDs [24]. More recently, low-grade intestinal inflammation, increased intestinal permeability, immune activation, and disturbances in the microbiome have been implicated [9]. Despite these recent discoveries, dysregulation of the gut–brain interaction remains the most important axis for understanding FGIDs. There is considerable evidence supporting the role of the central nervous system in the pathogenesis of FGIDs [22]. There are also studies that use heart rate variability in FGID patients to confirm dysfunction of the autonomic nervous system, indicating that suppressed vagal tone and overactive sympathetic tone are associated with FGIDs [25]. Yacob et al. found that 51.5% of children meeting the Diagnostic and Statistical Manual of Mental Disorders, fourth edition (DSM-IV), criteria for depressive or anxiety disorders had pain-predominant FGIDs [26]. A strong association between parental psychological status (particularly anxiety, depression, and somatization) and children’s abdominal symptoms has also been reported [27,28,29].

AIS is a condition where the spine undergoes three-dimensional deformity during adolescence without any specific cause [30]. The prevalence of AIS varies across the literature and has been reported to range from 0.47% to 5.2%. In our study, the overall prevalence of AIS over the 5-year period was 1.1% [31]. AIS is known to occur more frequently in females, with a reported incidence of 1.4 to 2.1 times higher than in males [31,32]. Consistent with existing knowledge, our study also found a higher incidence of AIS in females, with a 1.6-fold increase compared to males. AIS can lead to trunk deformity due to spinal distortion, and severe curvature may affect the function of the heart and lungs [33]. Surprisingly, there have been few reports regarding the impairment of gastrointestinal function and its association with digestive disorders within the abdominal cavity. The most well-known gastrointestinal disorder associated with AIS is superior mesenteric artery (SMA) syndrome. SMA syndrome occurs when the duodenum is compressed between the aorta and SMA, leading to symptoms of duodenal obstruction [34,35]. AIS is not a direct cause of SMA syndrome; however, SMA syndrome often occurs after corrective surgery for AIS. An increase in spinal length after AIS surgery is considered a risk factor, and other factors such as a lean body habitus, prolonged bed rest, and spinal orthoses are thought to influence its occurrence. Cases of SMA syndrome occurring after surgery in AIS patients with a lean body habitus are not uncommon in clinical practice, and numerous case reports exist [36,37].

Apart from gastrointestinal disorders related to surgery, there is currently no known association between common gastrointestinal disorders and AIS in the general AIS patient population. However, in clinical practice, we have encountered a considerable number of AIS patients who complain of non-specific gastrointestinal symptoms despite normal diagnostic tests. Therefore, we aimed to compare the proportion of patients finally diagnosed with FGIDs between the AIS group and the control group. Our goal was to investigate whether there is a correlation between AIS and FGIDs and explore the underlying factors.

In our multivariable analysis, FGIDs were more common in adolescents with AIS than in adolescents without AIS. It is generally accepted that adolescents with AIS have low self-esteem and poor mental health, possibly because of their trunk and chest deformities [11,12,13]. Furthermore, our previous large-database epidemiologic study revealed that the prevalence of psychiatric disorders was higher in adolescents with AIS than in those without AIS [15]. Considering the main etiology of FGIDs, we surmise that the psychological characteristics of adolescents with AIS are related to their high prevalence of FGIDs. Data from other interesting studies focusing on the relationship between psychological factors and FGIDs support this hypothesis. For example, excessive social media use has been strongly linked with psychological problems, such as depression and anxiety, and FGIDs [38,39]. A higher prevalence of FGIDs has also been reported in medical students, possibly because of the substantial stress imposed by medical curriculums [40,41].

As mentioned earlier, there are reports suggesting that FGIDs can occur when there is a limitation of movement due to conditions such as spinal cord injury [24]. While some AIS patients with severe scoliotic curves may experience a limitation of movement, such cases are very rare. Most AIS patients do not face significant challenges in their daily lives, except for cosmetic concerns related to body deformity. In this study, conditions with severe mobility impairment, such as neuromuscular scoliosis (M41.4), infantile and juvenile idiopathic scoliosis (M41.0, M41.1), and secondary scoliosis (M41.5) due to trauma, were all excluded from the AIS group based on diagnostic codes. Although there is a possibility of diagnostic code errors by some physicians, we believe such cases are extremely rare. In Korea, HIRA continuously monitors the appropriateness of diagnoses, and when doctors make misdiagnoses, they can face criminal prosecution and even imprisonment. Therefore, doctors are attentive to ensuring accurate diagnoses.

In a similar context, chronic gastrointestinal diseases such as peptic ulcer, Crohn’s disease, and ulcerative colitis may be associated with FGIDs. However, we used diagnostic codes that clearly diagnosed FGIDs, and the premise for entering such codes is the absence of other gastrointestinal diseases. Furthermore, even if other chronic gastrointestinal diseases coexist, their relevance to AIS has not been established, and we believe there will be no significant difference in the prevalence between the AIS group and the control group. Finally, if we were to set these diseases as exclusion criteria, it would raise the question of how many diseases should be included. Since our goal was a macroscopic comparative analysis using the HIRA dataset, a big data source, we aimed to keep the research design simple.

In this study, we focused on the known psychologic comorbidities in adolescents with AIS and found that these individuals were more likely to have FGIDs. Based on our experience as spinal surgeons, we interpreted the higher prevalence of FGIDs in AIS patients in our study more in relation to their mental health rather than their physical issues. We believe our findings will contribute to the better treatment of FGIDs in adolescents with AIS. When adolescents with AIS have functional abdominal symptoms, the surgeon can consider early referral to a pediatrician. In addition, if the spinal deformity is improved through corrective surgery, improvements in abdominal symptoms can be expected.

Our study has several limitations. First, we were unable to assess the impact of the severity of the scoliosis on the association between AIS and FGIDs because the severity information (e.g., Cobb angle or scoliosis treatment modality, such as surgery or bracing) was not included in the registry data. In particular, the past surgical history is an important factor that could be related to the manifestation of non-specific gastrointestinal symptoms. Unfortunately, due to the nature of the cross-sectional data in the HIRA dataset, we could not confirm this information. However, considering that a small proportion of adolescent patients undergo gastrointestinal surgery, we believe it would not have a significant impact on the results of our study. Also, we made efforts to account for a comprehensive range of other confounding factors, such as age, gender, insurance type, and residential district, making the results meaningful. Future research is needed to explore this further using datasets that provide more detailed information, including surgical history. Second, selection bias may be present. However, Korean National Health Insurance covers all citizens in Korea, and the percentage of individuals who never seek medical attention is exceptionally minimal. Thus, the datasets used in this study reflect the general population of our country. Third, the study patients were mainly Asian; so, it is uncertain whether our results would be generalizable to other population groups.

## 5. Conclusions

During the 5-year study period from 2012 to 2016, the prevalence of FGIDs was higher in adolescents with AIS (23.9%) than in adolescents without AIS (19.1%). In the multivariable analysis, where age, gender, insurance type, and residential district were taken into account, adolescents with AIS consistently exhibited a higher likelihood of FGIDs compared to those without AIS over the course of each of the 5 years. These results suggest that AIS may be a risk factor for FGIDs. It is advisable to provide counseling to adolescents with AIS and their parents regarding the heightened likelihood of experiencing persistent gastrointestinal symptoms. Surgeons should contemplate early referrals to pediatricians when such symptoms are evident.

## Figures and Tables

**Figure 1 children-11-00118-f001:**
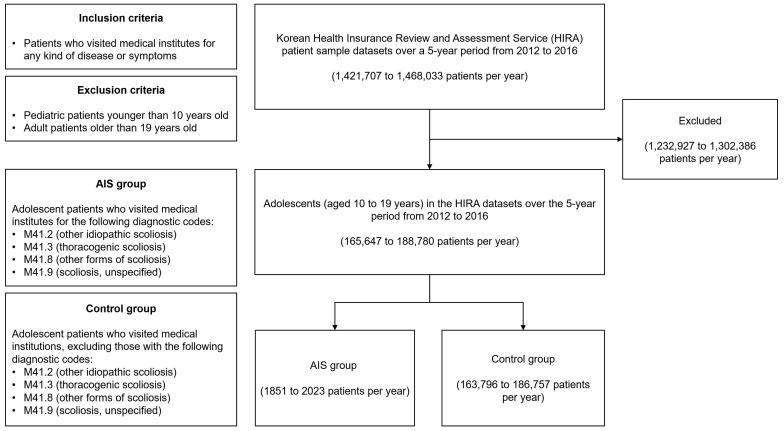
Flow diagram of the inclusion and exclusion criteria of the patients included in this study.

**Figure 2 children-11-00118-f002:**
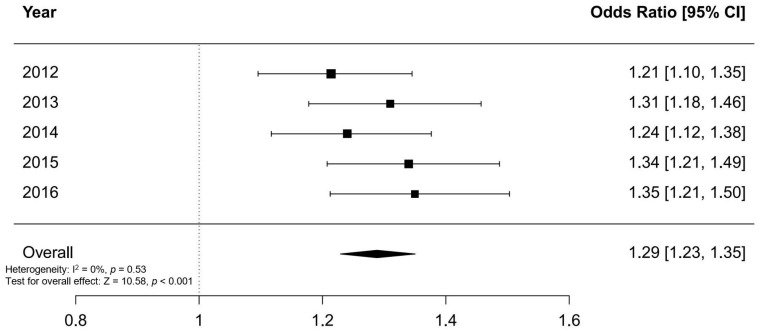
Forest plot showing the results of the fixed-effect meta-analysis on the association of FGIDs with AIS over the 5-year period. The overall odds ratio for FGIDs in adolescents with AIS was 1.29 (95% CI, 1.23–1.35).

**Table 1 children-11-00118-t001:** Prevalence and adjusted odds ratios of FGIDs in adolescents with AIS, compared with age-matched control adolescents, over a 5-year period.

Year	Prevalence in AIS Group, % (n *)	Prevalence in Control Group, % (n ^†^)	Adjusted Odds Ratio (95% CI)	*p* Value
2012	24.6 (498/2023)	21.0 (39,190/186,757)	1.21 (1.10–1.35)	<0.001
2013	24.9 (478/1918)	20.0 (36,099/180,183)	1.31 (1.18–1.46)	<0.001
2014	23.0 (460/2004)	19.1 (33,455/175,626)	1.24 (1.12–1.38)	<0.001
2015	23.9 (452/1892)	18.8 (31,655/168,653)	1.34 (1.21–1.49)	<0.001
2016	23.7 (438/1851)	18.5 (30,217/163,796)	1.35 (1.21–1.50)	<0.001

* number with FGID/total number with AIS. ^†^ number with FGID/total number without AIS.

**Table 2 children-11-00118-t002:** Multivariable logistic regression analysis of adolescents with AIS, compared with age-matched control adolescents, in 2015.

Variables	AIS (n = 1892)	Control (n = 168,653)	Adjusted Odds Ratio (95% CI)	*p* Value
FGIDs, % (n)				
Without FGID	76 (1440)	81 (136,998)		
With FGID	24 (452)	19 (31,655)	1.34 (1.21–1.49)	<0.001
Gender, % (n)				
Boys	38 (726)	51 (86,781)		
Girls	62 (1166)	49 (81,872)	1.69 (1.54–1.85)	<0.001
Age, % (n)				
10–12 y	21 (396)	29 (49,370)		
13–15 y	38 (721)	29 (48,266)	1.83 (1.62–2.07)	<0.001
16–19 y	41 (775)	42 (71,017)	1.33 (1.17–1.50)	<0.001
Insurance type, % (n)				
National health insurance	97 (1838)	96 (162,526)		
Medical aid	3 (54)	4 (6127)	0.75 (0.57–0.99)	0.040
Residential district, % (n)				
Metropolitan	24 (449)	16 (26,706)		
City	30 (576)	25 (41,569)	0.83 (0.73–0.94)	0.003
Rural	46 (867)	59 (100,375)	0.52 (0.46–0.58)	<0.001

## Data Availability

The data presented in this study are available on request from the corresponding author. The data are not publicly available due to HIRA policy.

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
