# Peer review of "Association of Functional Gastrointestinal Disorders with Adolescent Idiopathic Scoliosis"

_children, 2024, doi:10.3390/children11010118_

Round 1
Reviewer 1 Report
Comments and Suggestions for Authors
This article is an epidemiologic study of the association between adolescent idiopathic scoliosis (AIS) and functional gastrointestinal disorders (FGIDs) using disease registry data. The focus of the study is interesting, but the methodology needs to be revised, and the accompanying cases are not AIS and FGIDs in the medical sense of the term. Since this is a clinical study using disease registry data, it is fine as it is if it were submitted to a journal dealing with public health rather than a pathophysiology-oriented journal such as this journal. If the authors submit this paper to this journal, please address the following issues.
Problem 1: The authors included cases with the disease name AIS and excluded other cardinal diseases, but did the authors also exclude cases with underlying diseases with the disease name AIS? In other words, the accuracy of the disease name AIS needs to be improved in how this paper is written. Specifically, scoliosis secondary to cerebral palsy or trauma could be named AIS in the 2.2. study population (112-129), while M41.4 neuromuscular scoliosis and M41.5 other secondary scoliosis could not be named AIS in all cases. Not all cases are registered with a disease name. In order to exclude this risk, it is necessary to ensure that the disease of interest is not registered in the disease registry as cerebral palsy, trauma, or any other disease that can cause scoliosis.
Problem 2: FGIDs may include gastric emptying (duodenal folding) associated with scoliosis. Such cases are frequently encountered in daily practice. How did the authors confirm that such cases were not included in the FGIDs?
If we leave the ambiguities pointed out in issues 1 and 2, the odds ratios will naturally change.
Reviewer 2 Report
Comments and Suggestions for Authors
Evaluation report.
Thank you for sending me this manuscript for consideration. This study is well prepared and presented. I think that this study will make an important contribution to the literature. However, some revisions should be made.
1- Inclusion criteria should be determined (not clearly and unambiguously stated in the study)
2- Exclusion criteria should be determined (For example, those with GIS diseases such as peptic ulcer, Crohn's, ulcerative colitis with scoliosis should be excluded) (not clearly and unambiguously stated in the study)
3- Those who have undergone GIS surgery should be determined.
4- The relationship between scoliosis and GIS symptoms can be associated with limitation of movement. In such a case, individuals in the control group should not have limitation of movement and mainly gastro-intestinal system disease. (Nino-Murcia M, Friedland GW. Functional abnormalities of the gastrointestinal tract in patients with spinal cord injuries: evaluation with imaging procedures. AJR Am J Roentgenol. 1992 Feb;158(2):279-81. doi: 10.2214/ajr.158.2.1729781. PMID: 1729781.)
5- A detailed English revision of the manuscript is required.

Evaluation report.
Thank you for sending me this manuscript for consideration. This study is well prepared and presented. I think that this study will make an important contribution to the literature. However, some revisions should be made.
1- Inclusion criteria should be determined (not clearly and unambiguously stated in the study)
2- Exclusion criteria should be determined (For example, those with GIS diseases such as peptic ulcer, Crohn's, ulcerative colitis with scoliosis should be excluded) (not clearly and unambiguously stated in the study)
3- Those who have undergone GIS surgery should be determined.
4- The relationship between scoliosis and GIS symptoms can be associated with limitation of movement. In such a case, individuals in the control group should not have limitation of movement and mainly gastro-intestinal system disease. (Nino-Murcia M, Friedland GW. Functional abnormalities of the gastrointestinal tract in patients with spinal cord injuries: evaluation with imaging procedures. AJR Am J Roentgenol. 1992 Feb;158(2):279-81. doi: 10.2214/ajr.158.2.1729781. PMID: 1729781.)
5- A detailed English revision of the manuscript is required.
Reviewer 3 Report
Comments and Suggestions for Authors
The authors, through the analysis of a large sample of data collected over five years, concluded that FGIDs were more prevalent in adolescents with AIS compared to those without AIS. The study suggests that AIS may serve as a risk factor for FGIDs. The writing in the article is clear, and the analytical results are deemed reliable. Personally, I find this to be a commendable piece of work. I have a few minor suggestions, and it could be beneficial to present the analysis results in the form of images, which would enhance reader comprehension.
Round 2
Reviewer 1 Report
Comments and Suggestions for Authors
Regarding the ambiguity of the definition of AIS that we pointed out, a clear definition was described, and it became clear that this is a study population worthy of statistical consideration (121-122, 131-133,1255-266, Fig. 1). This paper is very readable and no longer feels uncomfortable. It may be published as is.
Reviewer 2 Report
Comments and Suggestions for Authors
The authors fulfilled their responsibilities. The study can be published.